# Molecular Detection of *Cryptosporidium* Species in Wildlife and Humans at the Wildlife-Human Interface around Queen Elizabeth National Park, Uganda

**Claire Mack Mugasa** [1,*], **Bernadette Basuta Mirembe** [2], **Sylvester Ochwo** [3], **Joseph Nkamwesiga** [3,4],
**Christian Ndekezi** [3,5], **Tobias Tusabe** [6], **Abubakar Musoba** [3] and **Clovice Kankya** [2]

1  Department of Biotechnical and Diagnostic Sciences, College of Veterinary Medicine, Animal Resources and Biosecurity, Makerere University, Kampala P.O. Box 7062, Uganda
2  Department of Biosecurity, Ecosystems and Public Health, College of Veterinary Medicine, Animal Resources and Biosecurity, Makerere University, Kampala P.O. Box 7062, Uganda
3  Molecular Biology Laboratory, College of Veterinary Medicine, Animal Resources and Biosecurity, Makerere University, Kampala P.O. Box 7062, Uganda
4  International Livestock Research Institute, Nairobi P.O. Box 30709-00100, Kenya
5  MRC/UVRI and LSHTM Uganda Research Unit, Plot 51-59 Nakiwogo Road, Entebbe P.O. Box 49, Uganda
6  Department of Pathology, Faculty of Medicine, Mbarara University of Science and Technology, Mbarara P.O. Box 1410, Uganda
*  Correspondence: claire1mack@mak.ac.ug

**Abstract:** To date, information on *Cryptosporidium* spp. infection status among people and wild animals living at the wildlife-human interface such as Queen Elizabeth National Park (QENP) is scarce. The aim of this study is to document the molecular detection of *Cryptosporidium* spp. in wild animals, and people, around QENP in the Kasese District. A total of 308 patients from four health centres and 252 wildlife animals from six species across 13 sampling areas were analysed microscopically and with PCR for *Cryptosporidium* spp. detection. The parasitological and molecular prevalence of *Cryptosporidium* spp. in humans was 40% and 53%, respectively; Kasenyi Health Centre recorded the highest percentage of positive stool samples for both tests. Wildlife species had an overall molecular percentage positivity of 30.16%; however, considering individual animal species that were sampled, the Waterbucks had the highest positivity rate, that is, 54.54%. All the samples were confirmed as genus *Cryptosporidium* with less species discrimination as our PCR target was a short fragment. There is a need to investigate the risk factors that predispose to high *Cryptosporidium* infection in the study area, especially in Kasenyi. In-depth investigation of the genetic diversity of *Cryptosporidium* spp. circulating at the human, livestock, and wildlife interface is imperative in devising disease management strategies.

**Keywords:** cryptosporidiosis; PCR; human-wildlife interface; Uganda

## 1. Introduction

*Cryptosporidium* is a Gregarine protozoan parasite of the order Eucoccidiida, suborder *Eimeriina*, and family *Cryptosporidiidae* [1]. Worldwide, *Cryptosporidium* is largely recognised as one of the most common diarrhoea-causing parasites [2,3]. It is an emerging zoonotic enteric pathogen, and its known zoonotic potential makes it a threat to public health worldwide [4]. The global importance of cryptosporidiosis as a diarrhoeal disease earned it a place on the World Health Organization's Neglected Diseases Initiative along with other infectious diseases [5]. In the last few decades, cryptosporidiosis has emerged as the second major cause of diarrhoea and death among infants in developing countries [6,7]. Environmental risk of *Cryptosporidium* spp. Infection, as for some other diarrhoeal diseases, has been linked to poor sanitation and hygiene and inadequate access to safe water with faecal contamination of food and water occurring in the environment [8,9]. The most common

clinical manifestation of cryptosporidiosis is diarrhoea which is characteristically profuse and watery and often contains mucus but rarely bloody or leucocytes [10,11]. Bowel motions may be abundant and as frequent as 10 or more per day, thereby contributing to rapid weight loss among patients [12,13]. Whereas *Cryptosporidium* infection usually manifests in acute self-limiting gastroenteritis, it can develop into a chronic and life-threatening diarrhoeal illness in immunocompromised individuals such as in HIV/AIDS [14].

The burden of cryptosporidiosis globally is measured as 8.4 million disability-adjusted life years (DALYs) and 99,800 deaths [15], whereas for children under 5 in Uganda, the total DALYs have been estimated to be 371,080 in 2016 [16]. Despite this burden, there are only a few studies about *Cryptosporidium* spp. infection and its burden in Uganda. Previous studies on *Cryptosporidium* spp. in Uganda have concentrated on areas with non-human primates [17–19], as well as in urban areas [20]. leaving a huge gap on *Cryptosporidium* infection among people living in more rural settings of Uganda as well as the infection status of people living at the wildlife interface, such as Queen Elizabeth National Park (QENP, https://doi.org/10.1101/2023.02.27.23286549, accessed on 30 January 2023), with more diverse herbivorous animals has been rarely reported [21]. Inquiry into the status of *Cryptosporidium* infection in rural areas close to livestock and wildlife will add valuable information such as the extent and prevalence of circulating species and probable management and control measures among others. The aim of this study, therefore, is to document the percentage positivity of *Cryptosporidium* infection among people attending selected health centres in the Kasese District, western Uganda as well as in selected wild animals in the same locality. The health centres act as sentinel sites for a catchment area of people living in the human-wildlife interface, bordering QENP.

## 2. Materials and Methods

### 2.1. Study Area

The study was carried out in the Kasese District that is located in western Uganda (0°12′ S and 0°26′ N; 29°42′ E and 30°18′ E), has an area of 3389.8 km$^2$, 86% is dry land, and 63% of the latter is for nature and wildlife conservation including Queen Elizabeth National Park (QENP), leaving limited land available for inhabitation and agriculture; 2% of the land area is occupied by permanent swamp and wetland [22,23]. Seventeen percent (17%) of QENP land is in the Kasese district and houses a number of fishing villages including Hamukungu, Kasenyi, Katunguru, and Katwe-Kabatoro (all in public enclaves), Kahendero, Katunguru, Kazinga, Kisenyi, Rwenshama, and Kayanja (park enclaves) (Figure 1). The study area thus commonly has wildlife, livestock and humans closely sharing this habitat as well as the natural resources such as water (Figure 2).

### 2.2. Study Population

The study participants who visited a health centre in the Kasese District with at least a trained nurse or laboratory technician (Katwe-Kabatoro HC, Karusandara, Kasenyi, Hamukungu, and Katunguru (Figure 1) and wild animals in selected areas within QENP (i,e., Kasenyi Koblek, Katunguru gate, Channel track, Janet track, Nyamunuka stretch, Camp site, Lake Katwe near institution, Research track, Katwe fishing village, Research koblek, Kaguta track, Mweya jet, and Mweya peninsula).

The community included in this study consisted of self-reporting patients at five health centre IIs in the Kasese District with diarrhoea and/or enteric complaints. Patient demographic characteristics including age, sex, HIV status, and village of origin were recorded.

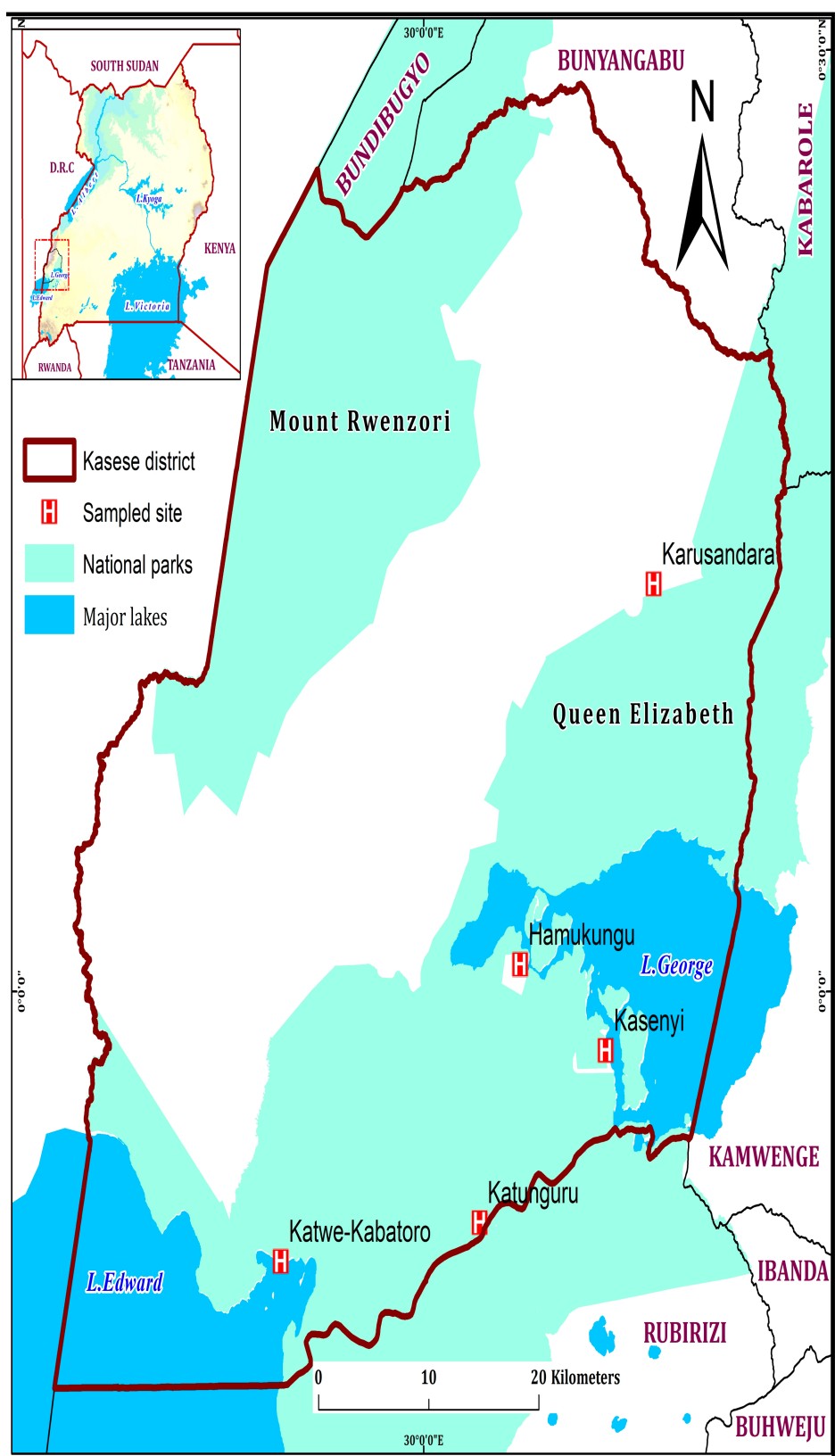

**Figure 1.** Study area showing the location of health centres (H) where study samples were collected.

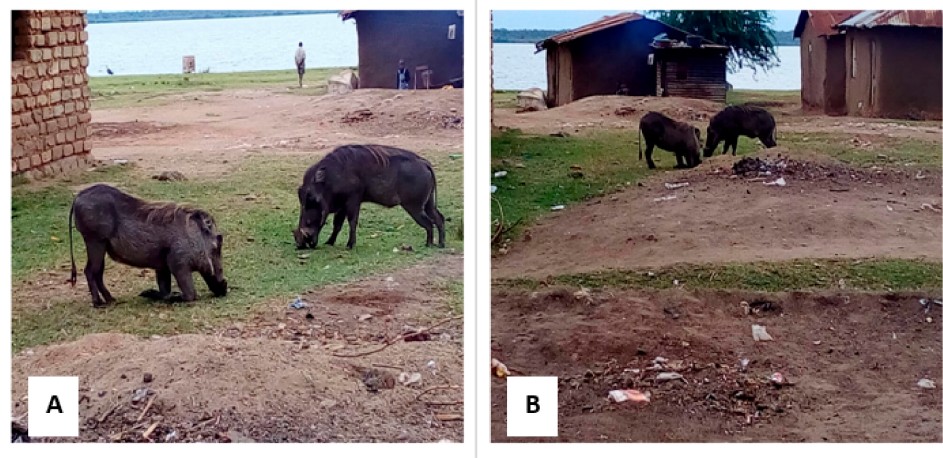

**Figure 2.** (**A**,**B**): Warthogs scavenging around homesteads in Kasenyi signifying a human-wildlife interface that is typical in the study area where faecal sampling was done.

### 2.3. Study Design and Sample Calculation

This was a cross-sectional study in which all consenting patients attending the five selected health centres with diarrhoea and/or enteric complaints and herbivorous wild animals around QENP. The samples were collected between June and October 2017. A total of 560 samples were collected, including 308 samples from symptomatic individuals in the community and 252 wildlife samples.

Exclusion Criteria

Patients that were moribund or those that did not consent.

### 2.4. Sample Collection, Processing, Transportation, and Storage

Approximately two teaspoonsful of stool samples were collected from patients attending health centre IIs in the study area. Individual patient's stool sample was collected into a screw-capped stool collecting container to which three parts of 75% ethanol was later added and mixed thoroughly. The containers were then further sealed with parafilm and coded before they were loaded into leak-proof biohazard bags and placed into sample carrier cool boxes ready for transportation to the Molecular Biology Laboratory (MOBILA), College of Veterinary Medicine, Animal Resources and Biosecurity (COVAB), Makerere University Kampala for laboratory analysis.

Fresh faecal samples of wild animals were collected by tracking selected wild herbivorous animals near and around the human settlements at the different landing sites around Queen Elizabeth National Park, Kasese District, Uganda. Briefly, herds of different target species were followed and forced to move to a safe distance to enable the researchers to access their fresh droppings. The faecal sample was similarly collected and treated as described in the human stool sample collection.

### 2.5. Microscopic Detection of Cryptosporidium Oocysts in Stool

Glass slides with air-dried thick faecal smears were placed on a staining rack, flooded with Auramine-phenol stain for 10 min, and then rinsed with running water. The smears were immersed briefly (2 min) in acid alcohol (0.5 concentrated HCL, 0.5 g NaCl, 75 mL of absolute ethanol, and 25 mL of distilled water), rinsed with running water, flooded with 0.1% of potassium permanganate for 30 s, rinsed with running water, and air dried. Slides were examined using fluorescence microscopy at low magnification ($\times$40). [24] *Cryptosporidium* oocysts appear as brightly fluorescent discs against a dark background (Figure 3).

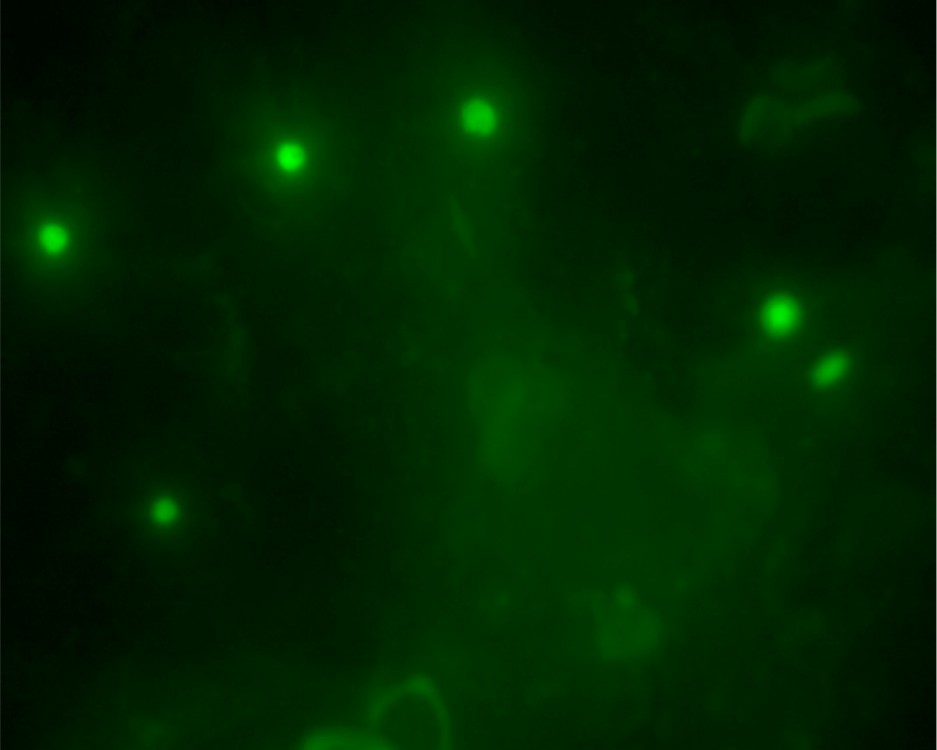

**Figure 3.** *Cryptosporidium* oocysts appear as fluorescent dots after staining with auramine-phenol.

Positive specimens were re-examined at high power magnification (×100) with oil immersion to ensure accurate identification of the oocysts. Three laboratory technologists performed the microscopic examination one being the tie-breaker.

### 2.6. DNA Extraction and PCR Amplification

DNA was extracted from all human samples and 100 wildlife samples randomly selected from the pool. Briefly, DNA was extracted from the stool samples using Quick-DNA Fecal/Soil Microbe Kits (Zymo Research, Tustin, CA, USA) with slight, in-house, modifications. All centrifugation steps were done at $10,000\times g$. Briefly, 900 µL of in-house prepared STE buffer (0.1 M Nacl, 10 mM Tris-Cl was added to 500 µL of faecal sludge in a 2000 µL (pH 8.0), 1% SDS, 1 mM NaEDTA (pH 8.0), 0.6 M β-Mercaptoethanol) microcentrifuge tube and incubated at 55 °C for 3 h with intermittent vortexing every 15 min. The microcentrifuge tubes were then centrifuged for 1 min, and 1000 µL of the supernatant was placed into a 2.5 mL collection tube to which 1000 µL of ice-cold absolute ethanol was added. The mixture (800 µL) was then transferred into a Zymo column™ (Zymo Research, Tustin, CA, USA) and centrifuged for 1 min until the entire mixture was passed through the column. DNA PreWash buffer™ (300 µL) was added to the column and centrifuged before washing with 500 µL of DNA wash buffer™. DNA was eluted in 100 µL of DNA elution buffer and stored at −20 °C until required.

PCR amplification: Molecular confirmation of *Cryptosporidium* spp. was carried out using a nested PCR targeting 18s rDNA as previously described by Kuzehkanan et al., (2011) [25], to yield an amplicon of 285 bp (Figure 4).

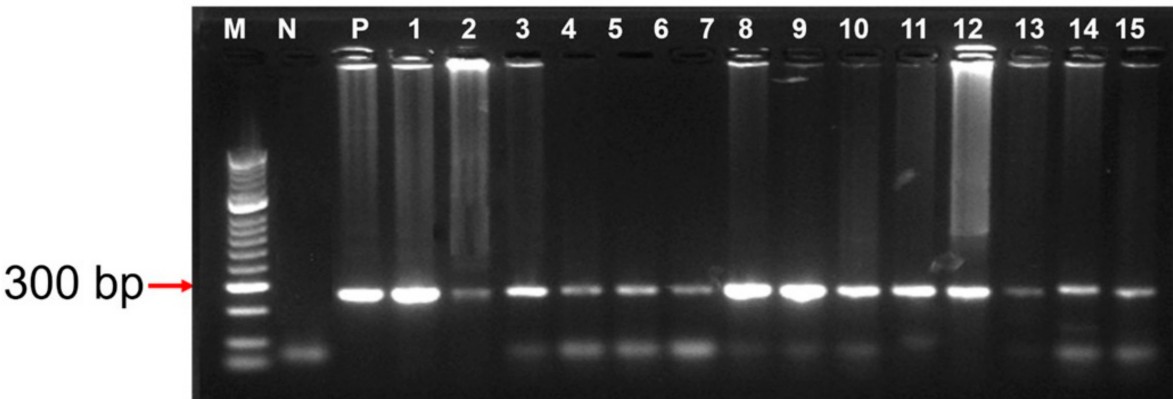

**Figure 4.** A selection of positive DNA samples after nested PCR; the arrow indicates the target band of 285 bp on the 100 bp molecular marker (M). N-negative control (Millipore water), P-positive control (previously confirmed with a COWP target: KY586963), 1–14 selected stool samples.

### 2.7. PCR Amplicon Sequencing

The PCR amplicons were purified using Qiagen PCR purification Kit (Qiagen, Germantown, MD, USA). We prepared 100 μL of the amplicons, 10 μL of the purified product was used to prepare representative gels, and the remainder was shipped to INQABA BIOTEC (Pretoria, South Africa) for sequencing. Purified PCR products were sequenced using the ABI 3500XL Genetic Analyzer, POP7™, BrilliantDye™ Terminator v3.1 (Thermo Fisher Scientific, Waltham, MA, USA). The nucleotide sequences of eight representative samples were deposited in the GenBank under accession numbers MK301300.1, MK301301.1, MK301302.1 MK301303.1, MK301304.1, MK301305.1, MK301306.1, and MK301307.1.

### 2.8. Data Analysis

The raw nucleotide sequences (ab1 files) were viewed and edited with BioEdit software version 7.0.0 [26], to remove any ambiguous sequences. Multiple sequence alignments and phylogenetic trees were constructed using MEGA X software (version 10) [27]. The representative reference sequences were downloaded from the web-based GenBank housed at the NCBI following a BLASTn search. The microscopy and PCR screening data were analyzed using STATA version 14 software to reveal the diagnostic accuracies of the two tests. Sensitivity was calculated by dividing the proportion of samples that tested positive divided by the total number of the true positives, whereas the specificity was calculated as true negative divided by the total number of samples that tested negative. Logistic regression analysis of various patient characteristics was completed using modified Poisson regression with robust standard errors to elucidate the possible associations. We computed the percentage prevalence of *Cryptosporidium* infection for both wildlife and human samples by dividing the number positive by the number tested. However, for wildlife samples, we did not have enough samples to categorically refer to the resulting proportion as "prevalence" since it may not be representative of the wildlife population in the entire national park.

### 2.9. Ethical Considerations

The study was reviewed by the School of Medicine, Research and Ethics Committee (SOM-REC) under the reference (#REC REF 2017-142) and approved by Uganda National Council for Science and Technology (UNCST) under review number HS 2333. The study was explained to each individual patient or guardian so as to attain informed consent. Following sample collection by the nurse or laboratory technician on call, the patients were treated for diarrhoea following the treatment regimen at the given health centres. No follow-up in the community was done because of budgetary constraints. Patients that met the inclusion criteria had their stool samples collected into sterile screw-capped stool collection containers.

### 3. Results

A total of 308 human faecal samples were collected from the five health centres. We analysed 282 samples in the laboratory since 8% (26/308) of the samples were either inaccurately collected or had their labels erased, thus excluded from the analysis. Most of the stool samples analysed were from Kasenyi HCII 28% (79/282), whereas the least were from Hamukungu HCII 13.5% (38/282). More than half of the faecal samples 54% (151/282) were collected from female patients. The majority of the patients, 31.6% (89/282), were between the ages of 5–18 years followed closely by children below five years of age, that is, 30.9% (87/282); the least number of samples (*n* = 6) were from patients over 65 years of age. The average age was 17 years and ranged from 3 months to 96 years of age. Watery stool samples accounted for 36.9% (104/282) of all samples analysed followed by solid hard stool, whereas soft solid stool accounted for the least proportion. (Table 1).

**Table 1.** Distribution of *Cryptosporidium* infection in stool samples from human patients around Queen Elizabeth National Park.

| Characteristic | Microscopy | | | PCR | |
| --- | --- | --- | --- | --- | --- |
| | Number Analyzed | Number Positive | % Positivity (95% CI) | Number Positive | % Positivity (95% CI) |
| **Location** | | | | | |
| Kasenyi HCII | 79 | 22 | 27.8 (18.3–39.1) | 19 | 24.1 (15.1–35) |
| Karusandara HCIII | 71 | 18 | 25.4 (15.8–37.1) | 13 | 18.3 (10.1–29.3) |
| Katwe-Kabatooro HCII | 49 | 8 | 16.3 (7.3–29.7) | 12 | 24.5 (13.3–38.9) |
| Katunguru HCII | 45 | 7 | 15.6 (6.5–29.5) | 11 | 24.4 (12.9–39.5) |
| Hamukungu HCII | 38 | 5 | 13.2 (4.4–28.1) | 3 | 7.9 (1.7–21.4) |
| **Age group (Years)** | | | | | |
| <5 | 87 | 27 | 31 (21.5–41.9) | 30 | 34.5 (24.6–45.4) |
| 5–18 | 89 | 28 | 31.5 (22–42.2) | 28 | 31.5 (22–42.2) |
| 19–35 | 70 | 17 | 24.3 (14.8–36) | 19 | 27.1 (17.2–39.1) |
| 36–65 | 28 | 3 | 10.7 (2.3–28.2) | 1 | 3.6 (0.1–18.3) |
| >65 | 6 | 0 | 0 (0–45.9) | 0 | 0 (0–45.9) |
| Not indicated | 2 | 0 | 0 (0–84.2) | 0 | 0 (0–84.2) |
| **Gender** | | | | | |
| Female | 151 | 81 | 53.6 (45.4–61.8) | 86 | 57 (48.7–65) |
| Male | 129 | 59 | 45.7 (36.9–54.7) | 55 | 42.6 (34–51.6) |
| Not indicated | 2 | 0 | 0 (0–84.2) | 0 | 0 (0–84.2) |
| **Stool consistency** | | | | | |
| Watery | 104 | 38 | 36.5 (27.3–46.6) | 38 | 36.5 (27.3–46.6) |
| Watery with rice pellets | 35 | 4 | 11.4 (3.2–26.7) | 4 | 11.4 (3.2–26.7) |
| Solid hard | 77 | 22 | 28.6 (18.8–40) | 22 | 28.6 (18.8–40) |
| Mucoid | 39 | 5 | 12.8 (4.3–27.4) | 5 | 12.8 (4.3–27.4) |
| Solid soft | 24 | 2 | 8.3 (1–27) | 3 | 12.5 (2.7–32.4) |
| Not indicated | 3 | 0 | 0 (0–70.8) | 0 | 0 (0–70.8) |

#### 3.1. Prevalence of Cryptosporidium Species

*Cryptosporidium parvum* was found to be the predominant circulating species in sampled patients. The overall prevalence of *Cryptosporidium* species in the community around QENP was 53% (95% CI 47–58) and 40% (95% CI 34–46) using microscopy thick smear and 18S rDNA nested PCR, respectively. Patients attending Kasenyi HCII recorded arguably the highest positivity (27.8% and 24% detected using microscopy and PCR respectively), whereas those attending Hamukungu HCII recorded the lowest microscopy and PCR positivity (13.5% and 8%, respectively) (Table 1). The watery stool showed the highest percentage positivity using either detection method; the soft solid stool showed the lowest positivity results when tested using microscopy, whereas the watery stool with rice pellets yielded the lowest percentage positivity upon testing with PCR.

The overall percentage positivity of *Cryptosporidium* among wild animals was 78% with warthogs showing the highest proportion of 86%. Sampling points with the highest positivity rates (100%) were Camp site 2 (*n* = 5), Channel track (*n* = 9), and Kasenyi Koblek *n* = 1), whereas Katwe fishing village had the lowest positivity rate of 40%. (2/5) (Table 2).

**Table 2.** Percentage positivity of *Cryptosporidium* spp. in wild herbivorous animals in QENP.

| Sampling Site | Wildlife Species Sampled (No. Positive/ No. Sampled) | | | | | | Overall |
|---|---|---|---|---|---|---|---|
| | **Buffalo** | **Elephant** | **Hippopotamus** | **Uganda Kob** | **Warthogs** | **Water Buck** | **% Positivity** |
| Camp site 2 | 0 | 100% (5/5) | 0 | 0 | 0 | 0 | 100% (5/5) |
| Channel track | 0 | 100% (5/5) | 100% (4/4) | 0 | 0 | 0 | 100% 9/9 |
| Janet track | 78% (7/9) | 0 | 0 | 0 | 0 | 0 | 78% (7/9) |
| Kaguta track | 0 | 0 | 0 | 0 | 100% (1/1) | 45% (5/11) | 50% (6/12) |
| Kasenyi Koblek | 0 | 0 | 0 | 100% (1/1) | 0 | 0 | 100% (1/1) |
| Katunguru gate | 0 | 0 | 0% (0/1) | 0 | 0% (0/1) | 100% (2/2) | 50% (2/4) |
| Katwe fishing village | 0 | 50% (1/2) | 0% (0/1) | 0 | 0 | 50% (1/2) | 40% (2/5) |
| Lake Katwe near institution | 0 | 67% (2/3) | 0 | 0 | 0 | 0 | 67% (2/3) |
| Mweya Jet | 67% (2/3) | 0 | 0 | 0 | 75% (3/4) | 0 | 71% (5/7) |
| Mweya peninsula | 0 | 0 | 0 | 100% (1/1) | 91% (20/22) | 0 | 91% (21/23) |
| Nyamunuka stretch | 0 | 67% (2/3) | 0 | 0 | 0 | 0 | 67% (2/3) |
| Research Koblek | 93% (14/15) | 0 | 0 | 0 | 0 | 0 | 93% (14/15) |
| Researchers track | 0 | 0 | 0 | 50% (2/4) | 0 | 0 | 50% (2/4) |
| **Overall** | **85% (23/27)** | **83% (15/18)** | **67% (4/6)** | **67% (4/6)** | **86% (24/28)** | **53% (8/15)** | **78% (78/100)** |

A total of 252 wild samples were analysed from different herbivorous animals within QENP, these include (Waterbucks, Uganda Kobs, Elephants, Hippopotamus, Buffalos, Warthogs). The majority of the samples were collected from buffalo (57 samples, 22.6%), whereas the least were collected from Hippopotamus (23 samples, 9.1%) (Table 2).

A blast search of the sequenced 18S rDNA showed that the query sequences (sequence from this study) were similar to *Cryptosporidium parvum* and *Cryptosporidium hominis*.

### 3.2. Microscopy Thick Smear Sensitivity and Specificity on Stool Samples

Microscopy thick smear sensitivity observed was 41.2% (95% CI 33.2–49.6), whereas its specificity was 61.2% (95% CI 52.4–69.5). A positive predictive value of 54.0% (95% CI 46.8–60.9) and a negative predictive value of 48.5% (95% CI 43.8–53.3) were observed while investigating human stool samples (Table 3).

**Table 3.** Comparison between of PCR and microscopy for the detection of *Cryptosporidium* in stool samples.

| | | Number of Samples with the Following Results Microscopy | | |
|---|---|---|---|---|
| | | **Positive** | **Negative** | **Total** |
| **PCR** | **Positive** | 61 | 52 | **113** |
| | **Negative** | 87 | 82 | **169** |
| **Total** | | **148** | **134** | **282** |

## 4. Discussion

In Uganda, *Cryptosporidium* infection has been shown to be an important risk factor for malnutrition including stunting in children, thus diagnosis and treatment of the infection may reduce the prospect of prolonged dehydration and the risk of wasting and probably loss of life [20]. The overall molecular prevalence of *Cryptosporidium* spp. in this study was found to be higher than what was previously reported (21%) within the vicinity of free-range gorillas in Bwindi Impenetrable National Park, Uganda [27] and among humans living around Kibale National Park also in western Uganda where prevalence was 32.4% [19]. This difference is probably associated with the current study area interfaces which are predominantly habituated by herbivorous wild game which are in close contact with humans. Proximity with livestock is known to be a risk factor for *Cryptosporidium* infection in humans [28,29]. In addition, the health centres sampled during our study are located close to open water bodies (Lake George and Lake Edward), thus patients live in communities in close contact with the water bodies. Proximity and fetching water from open water sources has been regarded as an epidemiological factor that increases the probability of *Cryptosporidium* infection [19,30,31] and may be one of the reasons for the high prevalence noted in this study.

The rural nature of the study area with limited health care coverage may also be a reason for the high prevalence of *Cryptosporidium* infection; this has also been reported in previous studies as a risk factor for infection [19]. Yet another study that was carried out in Gombe National Park, Kigoma District located in northwestern Tanzania revealed a much lower prevalence of 4.3% [32]. However, the population that was investigated in Kigoma was in close proximity with domestic animals specifically dog, goat, and/or sheep. Such a low prevalence may be explained by the varying potential of *Cryptosporidium* spp. to infect their hosts [33]. Elsewhere in Africa, the prevalence of *Cryptosporidium* has been reported in studies in various provinces in Egypt among children attending public hospitals; prevalence as high as 33.3% in the Ismailia province [34] and as low as 3.6% in Demietta province [35] have been recorded. Similarly, in South America, children younger than 2 years of age are frequently infected in developing countries like Peru [36] and Guatemala [37]. In the current study, children under five years had a molecular prevalence of 35%, slightly higher than the children aged 5–8 years (31%), although the number of stool samples was not sufficient to determine whether or not the difference was statistically significant.

A prevalence of 19.5%, much lower than that in the current study was reported in diarrhoeal patients of all age groups attending the outpatient clinics in Qaluibia province, Egypt [38]. The high prevalence registered in the current study could also be due to the use of water from sources contaminated by *Cryptosporidium* oocysts from faeces of livestock and wildlife. *Cryptosporidium* has been found to account for 60.3% of waterborne protozoan parasitic outbreaks globally from 2004 to 2010 [10]. The highest proportion of positive stool samples was recorded among patients attending Kasenyi Health Centre that is located in a busy Kasenyi fishing village and landing site. Such areas are characterised by a high population of people, limited toilet coverage, limited access to potable water, and poor general hygiene which are risk factors for *Cryptosporidium* infections and cryptosporidiosis [39–42]. It is thus crucial that genotyping is done in such areas to determine the level of genetic diversity of infecting *Cryptosporidium* spp., the source of contamination, and the route through which they are transmitted.

There are a number of molecular tools that have been employed in genotyping and subtyping; one of the frequently used tools is a gene that encodes a 60-kilodalton (KDa) precursor glycoproteins (GP60). Akiyoshi and others in 2006 utilised the GP60 gene to evaluate the genetic diversity of *Cryptosporidium* infection among children admitted to Mulago hospital, in Uganda. The data showed extensive genetic heterogeneity with three (3) new *C. parvum* alleles and suggested a high likelihood that the route of transmission is anthroponotic [43,44].

In the current study, sequencing of positive samples revealed that the infecting species were majorly *C. parvum*, and only one isolate was closer to *C. hominis*. The size of the 18s

rDNA PCR amplicon (285 bp) generated in this study, however, was not big enough to reveal major peculiarities among these sequenced isolates considering that the 18S gene is highly conserved among *Cryptosporidium* spp. The BLASTn search however revealed very high similarities with other species in the GenBank and thus formed the basis for their clustering and classification. Another target that has previously been employed in genotyping is *Cryptosporidium* oocyst wall protein (COWP) which was used by Nizeyi and colleagues in 2002 [28], who aimed to identify the genotype of *C. parvum* causing infections in humans sharing habitats with free-ranging mountain gorillas in the Bwindi Impenetrable National Park, Uganda [17]. Results from that study demonstrated *C. parvum* Genotype 2 infections in members of the local community living in the vicinity of the Bwindi Impenetrable National Park who frequently enter habitats of mountain gorillas and interact with them; the same genotype had been previously isolated from mountain gorillas [18] and particularly prevalent in cattle that grazed within the park [43], and elsewhere [45]. In previous studies elsewhere in Africa, *C. hominis* has been reported to be more dominant among infected humans unlike what is presented in the current study. In a study in the Ismailia province of Egypt, Helmy and colleagues in 2013 [46] reported that *C. hominis* accounted for 60.5% of human cryptosporidiosis compared to 38.2% of *C. parvum*. The prevalence was much higher than in the current study, probably because the previous study investigated samples from diarrhoeic children under 10 years old from farming households and in close proximity to animals, whereas the current study tested both children and adult patients. Another study in Nigeria by Ukwah et al., 2017 [47], reported that *C. hominis* was the most predominant species accounting for 48.0% of human cryptosporidiosis followed by *C. parvum* which accounted for 44.0%. These studies suggest that the transmission is anthroponotic. However, another study done in Ethiopia by Adamu et al., in 2010 [48] reported only one isolate as *C. hominis* compared to 39 out of 41 that were characterized as *C. parvum*; this is similar to the findings in our study where only one isolate was characterized as *C. hominis*, suggesting zoonotic transmission. In our study, the analysed faecal matter of wild animals revealed infection among all sampled animal species and culminating in a very high positivity rate of *Cryptosporidium* infection of up to 78%. This scenario increases the likelihood of *Cryptosporidium* spp. transmission to humans and their livestock through faecal contamination of water sources and vegetation by wildlife faecal contamination [30]. A total of 28 faecal samples of warthog were collected; the highest number (20/28) with a 90% positivity rate was collected from Mweya Peninsula, an area that is populated by tourists as well as local communities and especially employees of the various tourism establishments in QENP. Other notable wild animal species are buffalo (*n* = 27) and elephant (*n* = 18) which had appreciable positivity rates of 85% and 83%; in both cases, the positivity rates are much higher compared to that reported by Abu et al., 2011 in Kruger National Park (KNP), South Africa, which was 4.2% and 1.4%, respectively [49].

In the current study, microscopy was, in some instances, reported to detect more samples with *Cryptosporidium* infection than PCR. In our view, this may be attributed to the complex composition of faeces that may contain PCR inhibitors such as bile salts and polysaccharides. It has been reported that the presence of inhibitors can dramatically reduce the sensitivity and amplification efficiency of PCR [50]. Moreover, the use of auramine Phenol for staining the oocysts of *Cryptosporidium* in faecal samples prior to microscopy has been considered routine because it is inexpensive and fast. *Cryptosporidium* oocysts stained with auramine phenol are easily detected because they appear as very characteristic brightly fluorescent discs against a dark background using a fluorescent microscope. In their study to evaluate various tests in detecting intestinal cryptosporidiosis, auramine phenol was found to be simple fluorescent staining, highly sensitive, specific, cost-effective, and less time-consuming that can be relied upon for diagnosis [51].

Primarily, *C. parvum* is considered the zoonotic species of human cryptosporidiosis; infection in most cases is due to water contaminated from animal faeces and/or due to contact with animals [52]. There is, however, an indication that there is no true host

specificity of some species since they have been detected in human cryptosporidiosis, these species include *C. baileyi*, *C. canis*, *C. felis*, *C. meleagridis*, *C. bovis*, *C. suis*, *C. aAnderson*, and *C. muris* [47]. The zoonotic potential of these species has to be considered lower, although not in immunosuppressed persons [52].

There are reports in previous studies that human infection with *Cryptosporidium* has been by occupational exposure to infected animals (mainly calves) as well as companion animals, specifically dogs and cats; the latter have been implicated as probable sources of *Cryptosporidium* infection because they are sometimes infected with *Cryptosporidium parvum* [23].

This study shows for the first time that *Cryptosporidium* is circulating in populations in the QENP. The study also reveals a high prevalence of *Cryptosporidium* infection among patients attending health centres around QENP in the Kasese District as well as the high positivity rate among various herbivorous wild animals. However, in order to determine the zoonotic potential of *Cryptosporidium* spp. in QENP at the human/livestock/wildlife interface, prevalence and molecular characterisation of *Cryptosporidium* strains in patients and animals in communities neighbouring QENP should be carried out.

The study had limitations due to limited funding; genotyping the *C. parvum*-positives samples using markers such as gp60 that would have added value in relation to infection dynamics and genetic diversity of infection in the study populations was not done. The 18S rDNA amplicon that was sequenced in the current study was not long enough to generate differences usable in genotyping. In addition, the number of faecal samples as well as sites of sampling used may have been increased to better estimate the prevalence of infection in humans and animals.

There is a need to investigate the risk factors that predispose to such high *Cryptosporidium* infection in the study area, especially in Kasenyi where the prevalence was significantly higher than the rest of the areas studied. It is imperative to study the genetic diversity of *Cryptosporidium* spps in humans but also in livestock and wild animals using a longer PCR amplicon so as to generate more information on the species circulating and mode of transmission at the interface, thus devise strategies to control and management *Cryptosporidium* infection in the population.

**Author Contributions:** Conceptualization, C.M.M. and S.O.; Methodology. C.M.M., S.O., T.T., J.N., C.N., A.M. and B.B.M.; Software, C.N., J.N. and S.O.; Formal analysis, C.N., J.N. and S.O.; Investigation, C.M.M., B.B.M., S.O., T.T. and J.N.; Resources. C.M.M. and C.K.; data curation, C.M.M., C.N., J.N. and S.O.; writing—original draft preparation, C.M.M., S.O., C.K., J.N., C.N., A.M. and B.B.M.; writing—review and editing, C.M.M., S.O., T.T., C.K., J.N., C.N., A.M. and B.B.M.; visualization, C.M.M., J.N. and A.M.; supervision, C.M.M. and S.O.; project administration, C.K.; funding acquisition, C.K. All authors have read and agreed to the published version of the manuscript.

**Funding:** This research was funded by The Norwegian Programme for Capacity Development in Higher Education and Research for Development (NORHED), under the Project for Capacity Building in Zoonotic Disease Management Using Integrated Ecosystems Health Approach at the Human-Livestock-Wildlife Interface in the Eastern and Southern Africa (CAPAZOMANINTECO) project number UGA/0031/NORHED 1.

**Institutional Review Board Statement:** The study was conducted in accordance with the Declaration of Helsinki, and approved by the School of Medicine, Research and Ethics Committee (SOM-REC) of the College of Health Sciences, Makerere University under the reference (no. REC REF 2017-142 and approved by Uganda National Council for Science and Technology (UNCST) under review no. HS 2333.

**Informed Consent Statement:** Informed consent was obtained from all subjects involved in the study.

**Data Availability Statement:** Nucleotide sequences of eight representative samples were deposited in the GenBank.

**Acknowledgments:** The authors thank the health personnel at the study health centres in the study for enabling the collection of the stool samples, and Makerere University for facilitating the study.

Conflicts of Interest: The authors declare no conflict of interest.

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
