# Peer review of "Molecular Detection of Cryptosporidium Species in Wildlife and Humans at the Wildlife-Human Interface around Queen Elizabeth National Park, Uganda"

_parasitologia, doi:10.3390/parasitologia3020019_

Round 1

Author Response

Reviewer 1

  1. Firstly, the manuscript is in the mold of the periodical "Pathogens" and not "Parasitologia". Review.
    1. We have changed the skeleton of the manuscript to Parasitologia journal
  2. "Cryptosporidium" must be in italic
    1. We have italicized all the Cryptosporidium spps. Mentioned in the paper

  1. Line 18-20: The information "To-date, the Cryptosporidium infection status of people, wild animals living at the wild life-human interface such as Queen Elizabeth National Park (QENP) with more diverse herbivorous animals has not been reported." is not true. It can even be said that there are few records, but to say that this interface on cryptosporidiosis was not reported demonstrates the need for research in adequate databases. A quick search on PUBMED will find results on the analysis of this interface.
    1. We have changed the above statement as recommended by the reviewers. The new statement now reads as follow To-date, there is scanty information on Cryptosporidium spp. infection status among people and wild animals living at the wildlife-human interface such as: Queen Elizabeth National Park (QENP)” Line 18-20
  2. Replace "spp" with "spp."
    1. We have replaced all “spp” with “spp.” throughout the manuscript

  1. Line 24-25: The presentation of these percentages is confusing. I couldn't understand what is being expressed here, let alone how the p-value was arrived at (what is it comparing?).
    1. We have represented the results in table 1 and included confidence intervals. We have omitted the P-values since indeed there was no rational justification why we would use any one health center as a reference point for the rest. We have also reworded the title of the table to depict the results therein.

  1. Line 27: What is "highest positivity proportion of 54.54%"?

This line sentence has been revised to read more clearly as “…..however, considering individual animal species that were sampled, the Waterbucks had the highest positivity rate, that is 54.54%”. Lines 26-27

  1. Line 37: Suborder and family names should not be italicized.
    1. This has been effected in line 36

  1. Line 42: Reference number 5 is not properly presented.

Reference 5 has been properly presented      

  1. Line 61-62: It can even be said that there are few records, but to say that this interface on cryptosporidiosis was not reported demonstrates the need for research in adequate databases. A quick search on PUBMED will find results on the analysis of this interface. The information is not true.

  1. Thanks to the reviewers, this section has been updated and a citation on this regards was also added

  1. Line 82-83: The legend of Figure 1 must be described independently of the main text. Instead of a map of the study area, it should be a detailed description such as "Map of the study area of prevalence of Crypstosporidiosis in humans and wild animals during the period........ etc, etc, etc."

The legend has been edited to read “Study area showing the location of health centres (H) where study samples were collected. Line 134

  1. Line 85-86: The same in the caption of Figure 2. Information about the location of the animals in the photo is missing for it to be understandable regardless of the main text.

This has been revised to read “Warthogs scavenging around homesteads .in Kasenyi signifying a human-wildlife interface that is typical in the study area where faecal sampling was done”

  1. Line 205: As this is a Table legend, do not use abbreviated QENP.

This has been effected; QENP has been replaced with Queen Elizabeth National Park.

  1. The design of Table 1 and 2 could be improved.
    1. Table 1 and 2 have been improved on pg 7 and 8.
  2. The design of Table 3 is not a table, but a frame. Redesign it.
    1. This has been effected.

  1. Line 290-291: This is not a discussion, but a result.
    1. This section has been removed.

  1. I missed the discussion of the authors trying to explain why the PCR was less sensitive than the microscope analysis.

This has been discussed in Lines 558 to 566

Reviewer 2 Report

Dear authors,

Here are some considerations:

- Firstly, the manuscript is in the mold of the periodical "Pathogens" and not "Parasitologia". Review.

- Line 2: "Cryptosporidium" must be in italic.

- Line 18: "Cryptosporidium" must be in italic.

- Line 18-20: The information "To-date, the Cryptosporidium infection status of people, wild animals living at the wild life-human interface such as Queen Elizabeth National Park (QENP) with more diverse herbivorous animals has not been reported." is not true. It can even be said that there are few records, but to say that this interface on cryptosporidiosis was not reported demonstrates the need for research in adequate databases. A quick search on PUBMED will find results on the analysis of this interface.

- Line 21: "Cryptosporidium" must be in italic.

- Line 21: Replace "spp" with "spp."

- Line 23: "Cryptosporidium" must be in italic.

- Line 23: Replace "spp" with "spp." 

- Line 24: "Cryptosporidium" must be in italic.

- Line 24: Replace "spp" with "spp."

- Line 24-25: The presentation of these percentages is confusing. I couldn't understand what is being expressed here, let alone how the p-value was arrived at (what is it comparing?).

- Line 27: What is "highest positivity proportion of 54.54%"?

- Line 28: "Cryptosporidium" must be in italic.

- Line 29: "Cryptosporidium" must be in italic.

- Line 30: "Cryptosporidium" must be in italic.

- Line 31: Replace "spp" with "spp." 

- Line 37: Suborder and family names should not be italicized.

- Line 42: Reference number 5 is not properly presented.

- Line 44: Replace "spp" with "spp."

- Line 57: Replace "spp" with "spp."

- Line 58: Replace "spp" with "spp."

- Line 61-62: It can even be said that there are few records, but to say that this interface on cryptosporidiosis was not reported demonstrates the need for research in adequate databases. A quick search on PUBMED will find results on the analysis of this interface. The information is not true.

- Line 82-83: The legend of Figure 1 must be described independently of the main text. Instead of a map of the study area, it should be a detailed description such as "Map of the study area of prevalence of Crypstosporidiosis in humans and wild animals during the period........ etc, etc, etc."

- Line 85-86: The same in the caption of Figure 2. Information about the location of the animals in the photo is missing for it to be understandable regardless of the main text.

- Line 178: "Cryptosporidium" must be in italic.

- Line 205: As this is a Table legend, do not use abbreviated QENP.

- The design of Table 1 and 2 could be improved.

- The design of Table 3 is not a table, but a frame. Redesign it.

- Line 290-291: This is not a discussion, but a result.

- I missed the discussion of the authors trying to explain why the PCR was less sensitive than the microscope analysis.

Author Response

  1. Pg 1, Line 36 – Review the classification of the genus Cryptosporidium, which changed to “Gregarine protozoan”.
    1. This has been effected in line 36.

Materials and Methods

  1. Pg 4 Line 111 – Stool samples were transported in 75% ethanol. We know that ethanol is an inhibitor of the PCR reaction. Was the extraction method sufficient to remove this inhibitor? Could you talk a little about this in the discussion, could even explain the lower prevalence found in PCR compared to microscopy (Pg. 10, Line 242).

This has been discussed in Lines 558 to 566

  1. Pg 4 Line 125 – Add reference to the staining method.

Reference added Line 782

Results:

  1. The data in the tables must also be described in the text of the results.

This has been done Line 332-335 and 353-356

  1. 10, Line 228 Explain better in the text the calculation of sensitivity and specificity of the method

This has been effected in line 295-297

  1. It does not show the sequencing results, it is only mentioned in the discussion.
    1. Sequencing results have been added

Discussion

  1. When referring to prevalence in humans, it must be made clear that samples were collected from symptomatic individuals. (Pg. 4, Line 101, 102).

This has been clearly stated on pg 4, line 101

  1. Could you discuss the Auramine-phenol staining, since it is not the most used in Cryptosporidium detection papers.

This has been discussed, Line 616- 623

Round 2

Reviewer 1 Report

no comments

Author Response

Thank you for reviewing.

Reviewer 2 Report

Dear authors,

Excellent changes. Only two corrections still need to be made:

1) Throughout the text "spp." must not be italicized.

2) Tables must be constructed in the form of Tables and not Charts, containing only three horizontal lines and no vertical lines.

Kind Regards

Author Response

We have revised the manuscript according to the reviewer's comments. Please check the revised version.